# Bactericidal Activity of a Self-Biodegradable Lysine-Containing Dendrimer against Clinical Isolates of *Acinetobacter* Genus

**DOI:** 10.3390/ijms22147274

**Published:** 2021-07-06

**Authors:** Silvana Alfei, Debora Caviglia, Gabriella Piatti, Guendalina Zuccari, Anna Maria Schito

**Affiliations:** 1Department of Pharmacy, University of Genoa, Viale Cembrano, 16148 Genoa, Italy; zuccari@difar.unige.it; 2Department of Surgical Sciences and Integrated Diagnostics (DISC), University of Genoa, Viale Benedetto XV, 6, 16132 Genova, Italy; Cavigliad86@gmail.com (D.C.); gabriella.piatti@unige.it (G.P.); amschito@unige.it (A.M.S.)

**Keywords:** multi-drug resistant (MDR) clinical isolates, cationic lysine-modified dendrimer, self-biodegradability, *A. baumannii*, *A. pittii*, *A. ursingii*, *A. johnsonii*, MIC values (MICs), time-kill experiments, bactericidal activity

## Abstract

The genus *Acinetobacter* consists of Gram-negative obligate aerobic pathogens, including clinically relevant species, such as *A. baumannii*, which frequently cause hospital infections, affecting debilitated patients. The growing resistance to antimicrobial therapies shown by *A. baumannii* is reaching unacceptable levels in clinical practice, and there is growing concern that the serious conditions it causes may soon become incurable. New therapeutic possibilities are, therefore, urgently needed to circumvent this important problem. Synthetic cationic macromolecules, such as cationic antimicrobial peptides (AMPs), which act as membrane disrupters, could find application in these conditions. A lysine-modified cationic polyester-based dendrimer (G5-PDK), capable of electrostatically interacting with bacterial surfaces as AMPs do, has been synthesized and characterized here. Given its chemical structure, similar to that of a fifth-generation lysine containing dendrimer (G5K) with a different *core*, and previously found inactive against Gram-positive bacterial species and *Enterobacteriaceae*, the new G5-PDK was also ineffective on the species mentioned above. In contrast, it showed minimum inhibitory concentration values (MICs) lower than reported for several AMPs and other synthetic cationic compounds on *Acinetobacter* genus (3.2–12.7 µM). Time-kill experiments on *A. baumannii*, *A. pittii*, and *A. ursingii* ascertained the rapid bactericidal effects of G5-PDK, while subsequent bacterial regrowth supported its self-biodegradability.

## 1. Introduction

Gram-negative obligate aerobic coccobacilli belonging to the genus *Acinetobacter*, together with those belonging to the genera *Pseudomonas* and *Stenotrophomonas*, are non-fermenting pathogens responsible for serious nosocomial infections [1]. Particularly, bacteria of *A. baumannii* species, due to their increasing insensitivity to primary antimicrobial therapies, and their ability to rapidly adapt to the hospital environment, are becoming increasingly lethal for hospitalized immunocompromised patients [2]. Classified in the ESKAPE group pathogens, where ESKAPE is an acronym indicating the bacterial species *Enterococcus faecium*, *Staphylococcus aureus*, *Klebsiella pneumoniae*, *Acinetobacter baumannii*, *Pseudomonas aeruginosa*, and *Enterobacteriaceae*, the strains of *A. baumannii* resistant to carbapenems are considered by the World Health Organization the most important pathogens for which new therapies are urgently needed [3]. Experts in the field firmly believe that without significant intervention, hospital infections sustained by this pathogen will soon be incurable. In addition to *A. baumannii*, the most isolated hospital species of *Acinetobacter* include *A. nosocomialis* and *A. pittii*. These three species were grouped into *Acinetobacter calcoaceticus*–*A. baumannii* complex (Acb), which includes clinically relevant nosocomial pathogens. They are genetically closely related and phenotypically similar, but differ in epidemiology, antibiotic resistance, and pathogenicity [4]. In addition, *A. johnsonii* (usually found in the environment and in animals), *A. junii*, and *A. ursingii* are also characterized by a high tendency to develop drug resistance. They may occasionally colonize the skin of immunocompromised hospitalized patients, causing overt conditions, such as catheter-related bloodstream infections or peritonitis associated with peritoneal dialysis [5,6,7]. Although the mechanisms that support the ability of *A. baumannii* to overcome most therapeutic treatments have not been fully disclosed, this species more than others, has been shown to be insensitive to conventional antibiotic treatments.

The success of *A. baumannii* can be directly attributed to its flexible genome, which rapidly mutates when subjected to adverse conditions and stress. The production of β-lactamases and aminoglycoside-modifying enzymes, the reduced expression of outer membrane proteins (OMP), mutations in topoisomerases, and upregulation of efflux pumps, represent important mechanisms in determining antibiotic resistance in *A. baumannii* [8]. However, it has recently been reported that non-modified strains of *A. baumannii* differ from other Gram-negative bacteria, including *E. coli*, because of a different composition of lipid A. Indeed, in *A. baumannii*, the endotoxin of lipid A is a lipo-oligosaccharide (LOS) and not a typical lipo-polysaccharide (LPS), with hepta-acetylated C-11 alkyl chains in place of hexa-acetylated C-14 ones, and with an additional hydroxyl group [9]. Regarding the changes leading to the development of resistance, in the past, no adjustments in the external membrane charge (OM) like those developed by *P. aeruginosa* against colistin, were reported for *A. baumannii* [8]. On the contrary, more recently, colistin-resistant strains of *A. baumannii* have been detected, which, while maintaining the typical LOS structure of the wild type sensitive strains, modify the phosphate groups of lipid A, inserting galactosamine and/or phosphoethanolamine residues, reducing the net negative charge of the membrane and, thus, the susceptibility to AMPs, including colistin [9].

In addition to the described mechanisms of resistance, other fundamental mechanisms of virulence have recently been discovered, which allow *A. baumannii* and other clinically relevant species of this gender to thrive in the healthcare setting [10].

The clinical importance of *A. baumannii* resides not only on the wide range of infections it can cause in hospital settings, but also on those produced in certain types of communities. Conditions include skin and soft tissue involvement, urinary tract infections, meningitis, bacteremia, and pneumonia, the latter being the most frequently reported in both scenarios [11]. Sick patients forced into prolonged hospitalizations and characterized by immunosuppression, advanced age, the presence of comorbidities, trauma, or major burns, previous use of antibiotics, invasive procedures and the presence of indwelling catheters or mechanical ventilation, are candidates prone to develop nosocomial infections caused by *A. baumannii* [12,13,14,15]. Crude data on the mortality rates of patients following *A. baumannii* infections range from 23 to 68% [16]. Conversely, community-acquired infections are associated with a severe clinical syndrome that develop mainly in countries with warm, humid climates and typically occurs in individuals with other comorbidities, such as diabetes mellitus and chronic obstructive pulmonary disease, or in people who are heavy smokers or drinkers [11,17]. In this case, the reported mortality rate for community-acquired *A. baumannii* reaches 64% [18,19].

Hospital-acquired *A. baumannii* pneumonia is particularly associated with mechanically ventilated patients [20,21]. Generally, when caused by multi-drug resistant *A. baumannii*, the only successful therapeutic strategy is to use combinations of potent natural antibacterial peptides such as colistin, polymyxin B and tigecycline, the safety of which has not yet been well-documented [22,23,24]. Unfortunately, the accumulation of multiple mechanisms of resistance, including membrane modifications that allow *A. baumannii* to become insensitive even to colistin, could result in the development of “pan-resistant” strains.

In this alarming scenario, consisting of therapeutic options that are not entirely safe and whose activity will probably not endure, it is necessary to search for new antibacterial agents, active particularly against *Acinetobacter* and especially on *A. baumannii*. Natural AMPs have shown high potential to replace conventional and no longer effective antibiotics. Being highly susceptible to inactivation by peptidases, for years AMPs have inspired the synthesis of more stable antibacterial cationic materials, including stabilized peptides, enantiomeric peptides not susceptible to proteolysis, positively charged polymers, copolymers, and dendrimers [25,26,27,28,29,30,31]. Generally, synthetic macromolecules, such as polymers and dendrimers, compared to natural AMPs or low molecular weight (MW) peptides, have the advantages of possessing better antibacterial potency, a lower tendency to develop resistance, greater stability, long-term activity, and reduced toxicity [25]. Although these compounds may also act via specific mechanisms, such as inhibiting protein synthesis, DNA replication, the synthesis of cell wall, or inducing cell apoptosis and necrosis, the main recognized mechanism to explain their antimicrobial activity is an irreversible, non-receptor mediated, damage to microbial membranes, leading to lytic or non-lytic bacterial death [25,26,27,28,29,30,31].

In this study, a new fifth generation lysine-modified dendrimer (G5-PDK), having a highly cationic shell, essential for antibacterial activity, was synthetized. It was characterized by possessing a flexible difunctional propane-diol *core*, and a polyester-based biodegradable inner architecture. The results obtained from the physicochemical characterization showed that G5-PDK could possess high potential as antibacterial agent, and could be suitable for biomedical applications, in terms of surface charge, particle size, water-solubility, stability in aqueous solutions, and profile of protonation.

Preliminary experiments were conducted to determine the MICs of G5-PDK on a few strains representing relevant Gram-positive and Gram-negative species, most of them MDR. According to its structure, like that of G5K, a cationic dendrimer previously reported by us and specifically active on non-fermenting Gram-negative bacteria [28], also G5-PDK showed high MIC values on Gram-positive bacteria and on *Enterobacteriaceae*. Conversely, MIC values lower than those reported for several AMPs and other synthetic peptides were observed on nosocomial isolates of the genus *Acinetobacter*, including MDR strains of *A. baumannii*.

Considering the clinical relevance of *A. baumannii*, and the limited number of studies on the antibacterial activity of cationic dendrimers on this species, we decided to investigate in more depth the antibacterial effects of G5-PDK on clinical isolates of *A. baumannii*, and on other isolates of the genus *Acinetobacter*. Time-kill studies were also performed on representative MDR strains of *A. baumannii* and on isolates of *A. pittii* and *A. ursingii*, providing evidence of a rapid biocidal activity of G5PDK and of its self-biodegradability.

## 2. Results and Discussion

### 2.1. Uncharged Polyester-Based Dendrimer Inner Scaffold (G5-PD-OH)

G5-PD-OH was prepared through a long synthetic path, starting from the AB_2_ monomer 2,2-*bis*(hydroxymethyl)propanoic acid (*bis*-HMPA) [32], a known building block used to prepare a wide variety of dendrimer scaffolds. To this end, the synthetic strategy known as “double-stage convergent approach”, proposed in 1998 by Ihre and co-workers, was adopted [33]. Initially, we performed a series of selective protection and deprotection operations to prepare the fifth generation dendron D5-A-COOH (Figure 1).

The dendron growth was easily monitored through FTIR and NMR spectroscopy, which allowed us to verify the absence of defective intermediates dendrons, minimizing the risk of structural imperfections of the final dendrimer. Careful column chromatography was necessary for purifying the completely protected form of the intermediate dendrons and of G5-A-COOH from secondary products, such as anhydrides and *N*-acylureic derivatives, caused by side reactions of the activated acid reagents.

D5-A-COOH was covalently linked onto 1,3-propanediol (PD) *core*, obtaining the acetonide protected dendrimer G5-PD-A. After isolation, purification by column chromatography and characterization, G5-PD-A was deprotected with acid resins, providing G5-PD-OH as sticky resin (Scheme 1) [34]. For clarity, in the image reported in Scheme 1, only simplified representations of the real structures of the dendron and dendrimers were reported. Particularly, the acetonide groups in the structure of G5-PD-A are 32, while the peripheral OH groups in the structure of G5-PD-OH are 64. The same approach has also been used in editing Scheme 2 and Scheme 3.

G5-PD-OH was characterized by elemental analysis, and by FTIR, ^1^H and ^13^C NMR spectroscopy. The results were in perfect agreement with those reported in the literature [32]. G5-PD-OH was exploited to prepare the cationic dendrimer of this study.

### 2.2. Synthesis and Spectroscopic Characterization of the Polyester-Based Cationic Dendrimer (G5-PDK * 128 HCl)

#### 2.2.1. Synthesis and Spectroscopic Characterization of G5-PD-BK

To prepare the Boc-protected dendrimer G5-PD-BK, commercially available lysine protected with *tert*-butyloxycarbonyl group (Boc_2_-Lys-OH) was grafted onto G5-PD-OH through esterification of the 64 peripheral hydroxyl groups promoted by *N*-ethyl-*N*-(3-*N,N*-dimethylamino)propyl carbodiimide (EDC), in the presence of 4-(dimethylamino)pyridine (DMAP), as catalyst (Scheme 2). The reaction was conducted in dimethylformamide (DMF) for 24 h at room temperature (r.t.). The use of basic EDC required only a final extractive work-up after hydrolysis, and no further purification, to obtain G5-PD-BK analytically pure, being the ureic and acylureic by-products derived from EDC easily removable by acid washings.

Dendrimer G5-PD-BK was a glassy solid, soluble in almost all organic solvents except for pentane, hexane, cyclohexane, petroleum ether, and diethyl ether. FTIR spectrum of G5-PD-BK (Figure 2), was characterized by two strong bands, which have been evidenced in Figure 2 by two red circles. Particularly, at 1710 cm^−1^ it was detected the band of the C=O urethane groups, and at 1744 cm^−1^ that of the C=O ester groups, thus indicating the presence of the Boc groups and of the ester groups, respectively (Figure 2).

The MW of G5-PD-BK was computed on the base of the chemical structure established by the ^1^H NMR spectrum and was confirmed by elemental analysis. 

#### 2.2.2. Synthesis and Spectroscopic Characterization of G5-PDK * 128 HCl

The removal of Boc groups, to achieve the final cationic dendrimer G5-PDK as hydrochloride salt, was performed in acidic conditions with anhydrous HCl in situ produced by reacting acetyl chloride with methanol, conditions that have been shown to be compatible with the ester matrix of the dendrimer (Scheme 3). 

G5-PDK was firstly investigated by FTIR analysis to confirm the total removal of Boc groups, and the presence of the ammonium hydrochloride groups. As observable in Figure 3, where the FTIR spectrum of the dendrimer Boc-protected has been compared with the spectrum of cationic dendrimer, the C=O band at lower values of wavenumber, related to urethane carbonyl group of Boc (red circle), present in the spectrum of G5-PD-BK, was absent in the spectrum of G5-PDK, while a very broad and intense band in the range 3500–3000 cm^−1^, belonging to the 128 NH_3_^+^Cl^−^ groups, appeared (fuchsia circle). As desired, the C=O band at 1746 cm^−1^, related to esters carbonyl group, was maintained (fuchsia circles).

The ^1^H NMR spectra were very helpful both in following the chemical modifications, which occurred in the structure of dendrimers along the synthetic path from G5-PD-OH to G5-PDK, and for having confirmation of the chemical structures both of Boc-protected intermediate and of the final cationic dendrimer. In this regard, when G5-PD-OH was transformed into the Boc-protected dendrimer G5-PD-BK, containing the lysine residues, the broad signal of the 64 hydroxyl groups at 4.37 ppm and the complex signal of the 128 protons of methylene of the CH_2_OH groups at 3.98 ppm, typical of G5-PD-OH, disappeared, while new signals appeared belonging to the proton atoms of the Boc-protected lysine. Particularly, broad signals appeared in the range 0.95–1.90 ppm, where also the signals of the CH_3_ groups of the dendrimer were present. Additionally, two intense peaks were detected in the same region, and precisely at 1.43 and 1.44 ppm, belonging to the CH_3_ of the two types of *tert*-butyl groups. A broad multiplet, corresponding to the 128 protons of the CH_2_NHBoc of lysine was detectable at 3.10 ppm. Finally, very low and broad signals were observable in the range 4.70–5.50 ppm, relative to the NH-Boc groups. After the removal of the protecting groups, these last signals disappeared and, in the spectrum of G5-PDK, two intense signals belonging to the NH_3_^+^ cationic groups and integrable for 384 protons, popped up at 8.20 and 8.80 ppm.

The MW of the cationic dendrimer G5-PDK computed on the base of the chemical structure established by the ^1^H NMR spectrum, was confirmed by elemental analysis, and was experimentally validated by the volumetric titration of NH_3_^+^ groups.

### 2.3. MW Determination by Volumetric Titrations

To confirm the value of MW of G5-PDK obtained by ^1^H NMR analysis, and already validated with the elemental analysis, without resorting to routinely used, well-known, but very expensive techniques, such as MALDI-TOF, we carried out volumetric titrations. The already reported and validated method [35,36] consists in the titration of amine hydrochlorides with a standardized HClO_4_ solution in acetic acid (AcOH), in the presence of mercuric acetate and quinaldine red as indicator [37]. The procedure proved to be simple and reliable [35,36], its accuracy was secured by a sharp endpoint of titration, while its reliability was supported by the reproducibility of results. Table 1 collects, together with other physical features of G5-PDK, its calculated (NMR and elemental analysis), and experimentally observed MW (titration).

The very good agreement of observed value with the calculated one (error = −0.9%) confirmed the molecular structures of the prepared dendrimers and the goodness of the method.

### 2.4. Potentiometric Titration of G5-PDK

Our cationic dendrimer is characterized by not having quaternary and permanently protonated ammonium groups, but reversibly protonable primary amine groups depending on the pH value. Since a high degree of protonation is essential for a remarkable antibacterial activity and for a hypothetical clinical application of G5-PDK, it was important to know the pH values at which G5-PDK could be protonated and, above all, if it will be protonated in the physiological pH range of 4.5–7.5. To obtain this data, we performed a potentiometric titration of G5-PDK, according to Benns et al. [38]. Data of titration are reported in Table 2.

By reporting in the graph, the measured pH values vs. the aliquots of HCl added, the titration curve was obtained (Figure 4, red line, S.D. bars undetectable). Subsequently, from titration data, the dpH/dV values were computed and reported in Table 2. By reporting in graph these values vs. those of the volumes of HCl, the first derivative line of the titration curve was obtained (Figure 4, light blue line). The maximum of this curve corresponds to the volume of HCl, which allowed a compound in the protonated form. Interestingly, two points of maximum were observed, thus establishing the existence of a two-step protonation process. Data reported in the last three rows of Table 2, evidence that G5-PDK protonation of G5-PDK started at pH = 6.85, and total protonation occurred at pH = 4.80.

Non-fermentative bacteria are routinely reported for their ability to produce acids from six different carbohydrates (glucose, xylose, mannitol, lactose, sucrose, and maltose) by oxidative processes [39]. Therefore, in the case of bacterial colonization, the increase in bacterial biomass will produce an increase in CO_2_ and acids deriving from the oxidation of D-glucose, thus causing a slight decrease in pH, even in districts where in normal conditions the pH is higher than 6.85. The consequent decrease in pH will ensure efficient protonation to the dendrimer, thus favoring the electrostatic interactions of the device with the negatively charged bacterial surface.

### 2.5. Dynamic Light Scattering Analysis

Figure 5 shows the particles size distribution related to one of the three determinations performed, reported as number-weighted size distribution (**a**) and as intensity-weighted size distribution (**b**).

The mean particle size of G5-PDK (203.0 nm) and the mean PDI (0.282) were superimposable both considering the intensity-based results and the number-based ones. Note that, the only real measured value through DLS is the scattering intensity, while the other quantities, such as number and volume, are derived using different equations, in which invariably, several assumptions are made, which could (or not) introduce a large bias, depending on the PDI of the sample. In our case, to have obtained identical mean hydrodynamic diameters, both considering the intensity-based results and the number-based ones, establishes that the population of G5-PDK particles is very homogeneous. The ζ-p was positive and precisely the values was +19.2 mV (Figure 6).

It is known that small particles assure minor tissue toxicity, but extremely minute particles could easily undergo hepatobiliary and renal clearance [40]. A correct balance to minimize both tissue toxicity and fast clearance by the mononuclear phagocytic system (MPS) is the best solution [41,42]. Although the size of nanoparticles should not be above 100 nm to reach, for example, tumor tissues by passing through vascular structures, more, in general, 20–200 nm particles could have the highest potential for in vivo applications [43], thus establishing the possible use of G5-PDK for clinical conditions. The surface charge of G5-PDK (+19 mV) was sufficiently high to avoid the tendency to form aggregates over time in solution and adequately positive to ensure the cationic surface essential for interacting with the bacterial outer envelope and for damaging bacterial membranes. Indeed, G5-PDK displayed a high solubility in water, providing solutions stable over time.

### 2.6. Antibacterial Activity of G5-PDK

#### 2.6.1. Design of the Structure of G5-PDK

The structure of G5-PDK was designed based on the recent results obtained by us concerning cationic copolymers and cationic amino acids-modified polyester-based dendrimers [27,28,29,31]. We observed significant differences between the antibacterial activity of the cationic copolymers [29,31] and that of the cationic dendrimers [27,28]. While cationic copolymers showed broad-spectrum activity against most of the 61 tested MDR isolates [29,31], belonging to both Gram-positive and Gram-negative species, cationic dendrimers showed high specificity for some families of bacteria. We assumed that the random and not-controlled structure of copolymers, and the cationic charge randomly distributed along the whole polymer backbone, could translate in a higher multivalency, associated to a higher adaptability (intended as capability to interact with) to different types of bacterial surfaces. On the contrary, the highly precise tree-like architecture typical of dendrimers, associated with the uniquely peripheral and extremely ordered position of cationic amino acids, could have reduced their multivalence, and decreased their adaptability to surfaces of a limited category of bacteria. Moreover, the type of amino acid used to functionalize the uncharged scaffolds of dendrimers influenced their activity. As for the three previously synthetized fifth-generation amino acids-modified dendrimers, characterized by having 192 protonated nitrogen atoms (G5K, G5HK and G5H), they were inactive against Gram-positive bacteria and *Enterobacteriaceae*, but were specifically active against Gram-negative non-fermenting species [28]. Lysine proved to be essential for improving their potency, while histidine was not helpful and therefore no longer considered in our successive research. By associating lysine with arginine, whose guanidine group is known to help redirect the activity of antibacterial cationic macromolecules to Gram-positive species, we obtained arginine modified G4 and G5 dendrimers that were active only on MDR isolates of Gram-positive species [27]. Among them, the lysine/arginine dendrimer G5KR, characterized by the largest number of cationic groups, was found to be the most active. The failed production of a fourth-generation dendrimer containing lysine (G4K), a lower generation counterpart of the powerful antibacterial dendrimer G5K with half of the cationic groups (96), confirmed the importance of a high positive charge density for antibacterial power (data not reported).

Based on these considerations, with the aim of obtaining a dendrimer with different antibacterial effects than those previously reported, we decided to prepare a new dendrimer with an uncharged polyester matrix structured differently, and containing a different number of lysine, essential to have a powerful activity. This new dendrimer should have owned cationic groups in a number lower than 192 (as G5K, G5HK, and G5H), and higher than 96, which made G4K, ineffective. To this end, we prepared the uncharged polyester scaffold of G5-PD-OH, made of two dendrons of fifth generation esterified on a bivalent propanediol (PD) *core*, and possessing a total of 64 OH. After esterification with lysine, we would have obtained a positively charged dendrimer with 128 cationic groups. In addition, the presence of a linear *core* such as PD, instead of the branched triol present in previous dendrimers, would have provided a more flexible structure with different possibilities of interaction with the bacterial surface, thus favoring a different antibacterial activity.

#### 2.6.2. MIC Values Displayed by G5-PDK

The antibacterial activity of G5-PDK was firstly screened determining the MIC values on relevant representative of both Gram-positive and Gram-negative species, including MDR isolates (Table 3).

As predicted by its structure, resembling that of G5K, active exclusively on non-fermenting Gram-negative bacteria [28], G5-PDK displayed MICs > 25.4 µM on Gram-positive bacteria and *Enterobacteriaceae*. Although in several studies, such as that of Stenström et al. [44], MIC values of 100 μM were considered indicative of significant antibacterial activity, in our opinion values > of 25.4 μM were already too high to make consider a compound as active. We therefore deduced that these species were not susceptible to G5-PDK (Table 3). On the contrary, G5-PDK displayed MIC values far lower than those reported for several AMPs, against the *A. baumannii* clinical isolate tested in these early experiments (Table 3). Based on the clinical relevance of *A. baumannii* MDR strains, and the limited number of studies on the antibacterial effects of cationic dendrimers on this species, in this work we analyzed in detail relevant clinical isolates belonging to this specific genus. Accordingly, MICs for G5-PDK were obtained analyzing a total of 12 strains of clinical origin, belonging to different species of the *Acinetobacter* genus, which could be responsible for severe human infections (Table 4). The MICs observed for G5-PDK against the MDR isolates tested in this study were compared to the MICs obtained for ciprofloxacin, an antibiotic commonly used in clinical practice to treat infections caused by these pathogens (Table 4).

As can be seen in Table 4, G5-PDK was active against various strains of different species of the genus *Acinetobacter*, showing MIC values in the range of 6.3–12.7 µM on *A. baumannii* and MIC = 3.2–12.7 µM on other *Acinetobacter* species. Furthermore, the data reported in Table 4 show that G5-PDK can be considered an efficient antibacterial agent, capable of inhibiting numerous isolates of the genus *Acinetobacter*, also including those towards which the currently most used antibiotic ciprofloxacin is no longer active (MICs = 48.3–193.2 µM).

Note that the available studies on cationic materials tested against *A. baumannii* in recent years are very limited. Furthermore, the reported data are often conflicting, even when the same cationic antimicrobial compounds were tested on the same ATCC 19606 strain of *A. baumannii* [45,46]. This is the case of some AMPs reported by Giacometti et al. [45] and subsequently by Vila-Farres et al. [46]. In those studies, while Cecropin P1, Indolicidin, and Magainin II were initially shown to have MICs in the ranges of 0.1–9.6 µM, 1–33.6 µM, and 0.2–6.5 µM, with no distinction between 12 clinical isolates and an ATCC 19606 strain of *A. baumannii* [45]. In the second article, only single MIC values were reported, and differences in these values were observed between ATCC 19606 and clinical isolates when exposed to Indolicidin. On *A. baumannii* ATCC 19606, MICs were 0.5 µM for Cecropin P1 and 4.2 µM for Indolicidin (thus, within the previously reported ranges), but a MIC value of 103.8 µM was reported for Magainin II [46] (16-519-fold higher than the MICs previously reported). Regarding clinical isolates, only data on the antibacterial activity of Indolicidin were available, which established that Indolicidin was less effective on clinical isolates (MIC = 32 µM) than on ATCC 19606 strain (MIC = 4.2 µM) [46]. Based on these considerations, a reliable comparison of the antibacterial activity of G5-PDK with that of the previously reported AMPs could be problematic. Referring to the most recent study by Vila-Farres et al. [46], G5-PDK was much more active than several AMPs tested on *Acinetobacter* ATCC 19606.

G5-PDK was 3.4–6.8-fold more active than Bactenecin, 4.7–9.4-fold more than Bofarin 1, over 6.6-fold more effective than Histatin 5, 1.6–3.2-fold more than Histatin 8, 1.2–2.4-fold more than HNP-1, 2, 2.1–4.2-fold more than Magainin 1, and even 8.2–16.4-fold more than Magainin 2. β-Defensin, which, in the same experiment, displayed MICs = 65.6 µM was 5.2–10.4-fold less active than G5-PDK, which in turn proved antibacterial effects comparable to those of Cecropin A, B, and Indolicidin [46].

Even more interesting results were obtained when the antibacterial effects of G5-PDK reported by us were compared with those obtained by the Vila-Farres group, which tested Indolicidin, Colistin, and Mastoparan on clinical isolates of the genus *Acinetobacter* (susceptible to colistin), as in the present work. Indeed, G5-PDK showed MICs 1.3–2.6-fold lower than those of Indolicidin and slightly higher than those of the powerful Mastoparan (6.3 µM vs 5.4 µM) [46].

Moreover, G5-PDK was more active than 4 out of 7 cationic peptides tested by Jaśkiewicz et al. on *A. baumannii* ATCC 19606 [47]. Specifically, G5-PDK was 1.4–2.8-fold more potent than Omiganan and 7.2–14.4-fold than Temporin A.

As for synthetic peptides, G5-PDK was much more potent than the all-D-enantiomer antimicrobial peptidomimetic _D_(KLAKLAK)_2_, prepared by McGrath et al. to limit proteolysis, which is the main concern associated with the in vivo use of antimicrobial peptides [48]. The authors tested _D_(KLAKLAK)_2_ on different Gram-negative pathogens, also including *A. baumannii* ATCC 19606, reporting on this strain a MIC of 194.6 µM, thus showing a potency 15.3–30.6-fold lower than that of G5-PDK.

Recently, the Sharma’s group prepared a cationic peptide (SA4) and a cationic peptoid (SPO), which showed MIC = 65.1 µM and 132.8 µM, respectively, against *A. baumannii* ATCC 19606, and MIC = 32.6 µM and 66.4 µM, respectively, on four isolates of MDR *A. baumannii*. Consequently, G5-PDK tested against MDR clinical isolates of *A. baumannii* was 2.6–5.1-fold more active than SA4 and even 5.2–10.4-fold than the SPO [49].

Limiting our analysis to existing studies on cationic dendrimers with antibacterial activity against *A. baumannii*, the most interesting compound is João Pires’ synthetic dendrimer peptide G3KL [50]. In their work, the authors, unlike most studies that have used *A. baumannii* ATCC, tested G3KL against 32 clinical isolates of *A. baumannii* (including 10 OXA-23 isolates, 7 OXA-24 strains, and 11 OXA-58 I carbapenemase producing isolates) observing very low MICs, in the range 0.8–3.2 µM.

However, as far as we know, in the field of synthetic cationic dendrimers endowed with antibacterial activity on *A. baumannii*, G5-PDK is the one that shows MIC values closest to those of G3KL [50].

In this study, to investigate as much as possible the spectrum of activity on the genus *Acinetobacter*, we also considered other species beyond *A. baumannii*, such as *A. johnsonii*, *A. junii*, *A. pittii*, and *A. ursingii*. On these species, G5-PDK produced MICs in the same range, except for *A. ursingii* (strain 388) on which a lower MIC, equal to 3.2 µM, was observed. The comparison between the antibacterial activity of G5-PDK and that of agents previously reported against these different species has been impossible because, to the best of our knowledge, there is currently no study in which strains of such species have been tested against cationic materials.

#### 2.6.3. Time-Kill Curves

Time-kill experiments were performed with G5-PDK at concentrations equal to 4 × MIC on selected strains of *A. baumannii*, *A. pittii*, and *A. ursingii*. According to the results, G5-PDK showed a strong bactericidal effect against all pathogens. Indeed, a reduction of 5 logs in the original cell number was observable, after 2h, for *A. baumannii* 245 and *A. pittii* 272, after 4 h for *A. baumannii* 279 and 383, while for *A. ursingii* 408 a 4-log reduction was observed after 2 h of exposure to G5-PDK. Particularly, Figure 7 shows the time-kill curves obtained on *A. baumannii* 245, 279 and 383, for which the observed MICs were of 6.3 (strains 245 and 383) and 12.7 µM (strain 279). During the 24 h, a regrow was observed for all strains. A regrowth was detected starting from 4 h for *A. baumannii* 279, while no regrowth was detected for *A. baumannii* 245 and 383 for a period between 2 and 6 h. For these strains, regrowth began after at least six hours of exposure to G5PDK (Figure 7). Concerning *A. pittii* and *A. ursingii*, although a regrowth was evident during time-kill experiments, it was less marked if compared to that of isolates of *A. baumannii* species (data not shown).

Interestingly, the mechanism of action we observed here, overlapped that of the cationic dendrimer G5K, characterized by a similar polyester-based uncharged inner matrix peripherally esterified with lysine, which we recently reported [28].

On the contrary, for the polystyrene-based copolymers P5 and P7 not containing ester groups [29,31], different kinetics of action were observed, characterized by the absence of regrowth of the analyzed strains.

The main mechanism of action of cationic materials is based on the fact that non-permanently cationic macromolecules, such as G5-PDK, can exert rapid bactericidal activity only when protonated. The first protonation for G5-PDK occurs at pH = 6.85, which is lower than the pH of the culture broth (which is about 7.0–7.3). However, during the time in culture and the consequent increase in the biomass of *A. baumannii*, a slight decrease in pH could be explained as follows.

Non-fermenting gram-negative bacteria metabolize glucose using aerobic respiration and therefore produce a small number of weak acids during the Krebs cycle and Entner Doudoroff glycolysis [51,52]. By means of oxidative processes, they can produce acids from six carbohydrates (glucose, xylose, mannitol, lactose, sucrose, and maltose) [39], thus causing a lowering of the pH and guaranteeing the protonation of G5-PDK. When all the bacteria die, the production of acids ceases, thus causing an increase in pH to the initial values. The bacterial regrowth observed with polyester dendrimers, under these pH conditions, can be explained by assuming a process of self-degradation of the lysine-modified polyester structure of G5-PDK, which would lead to its inactivation, as reported [53,54,55,56,57]. The Mizutani group has in fact demonstrated that, in aqueous solution (pH > 7.0) at 37 °C, the simultaneous presence at the right distance in the macromolecules of ester groups, and non-protonated primary amino groups, can cause their self-degradation, thanks to an intramolecular amidation process [53]. In G5-PDK, as in the previously reported G5K, the ε amine groups of lysine moieties, which according to the protonation profile discussed in Section 2.4, should not be protonated at pH = 7.0–7.3 (first protonation occurs at pH = 6.85), may react with the ester carbonyl separated by 5 methylene groups, leading to the intramolecular cyclization and detachment of lysine residues, according to Scheme 4.

These events, leading to the progressive elimination of cationic groups from the dendrimer, cause the radical reduction of antibacterial activity with consequent regrowth of pathogens. Note that prior to Mizutani, a similar self-degradation mechanism had already been reported for polyester platforms containing primary amine groups in the side chains [54,55,56]. Furthermore, it has also been reported that intramolecular condensation of the terminal amino group with the carboxyl group of lysine esters produces a seven-membered α-amino-ε-caprolactam (2-aminohexane-6-lactam) [57]. The self-degradation strategy that uses groups of esters appropriately spaced from the primary amine inside chains would be useful to quickly reduce in vivo the antimicrobial effects responsible for the potential toxic side effects of cationic molecules observed on eukaryotic cells.

It should be noted that cases are rarely reported in the literature in which bactericidal behaviors of cationic materials towards various bacterial species lasted up to 24 h. Indeed, for cationic polymers, dendrimers and peptides, time-kill experiments up to 6 or 8 h of exposure have often been reported [58,59,60]. Furthermore, when for cationic bactericidal peptides, such as colistin [61] and dendrimers [28], the data are reported at 24 h; it can be observed that, while a rapid killing occurred even after only 5 min [61], and 1 h [28] from the contact with the cationic antibacterial device, on the contrary, after 24 h of action, an abundant bacterial regrowth happened. Regarding the cationic antibacterial agents tested specifically against *A. baumannii*, McGratha et al. reported the bactericidal behavior of a synthetic peptide by running time-kill experiments for 24 h and using concentrations equal to MIC, 2 × MIC and 4 × MIC. While at 2 × MIC and 4 × MIC, an 8-log reduction was obtained after 10 and 4 h of exposure to the antibacterial agent, and no further regrowth was observed up to 24 h, at the MIC value, the 8-log reduction was reached after 10 h, with a subsequent regrowth in the following period, as in our case [48]. Although the antibacterial potency of the cationic peptide seems notable; however, the concentrations necessary to determine the complete and definitive reduction of a high bacterial inoculum were extremely high (2 × MIC = 389.2 µM and 4 × MIC = 778.4 µM) [48]. In the case of the use of a relatively lower concentration (194.6 µM), complete bacterial elimination occurred after 10 h and regrowth was observed during the 24 h. In this regard, G5-PDK, which killed all bacteria after only 2–4 h at concentration of 4 × MIC [25.4 µM (strain 245) and 50.8 µM (strain 279)] was 3.8–7.7-fold more potent. Against *A. baumannii*, a more potent bactericidal agent was found to be the amino acid conjugated polymer containing glycine ACP-I (Gly) prepared by Barman et al. [58], but it was not possible to verify the regrowth of the bacteria because the time-kill experiments were only performed for up to six hours. It would also be interesting to compare the bactericidal activity of G5-PDK with the very powerful cationic antibacterial dendrimer peptide G3KL [50], but as in many studies on cationic antibacterial agents, no time-kill experiments have been performed.

#### 2.6.4. Authors Consideration about the Antibacterial Activity of G5-PDK

The results of this study highlighted that G5-PDK had a peculiar antimicrobial activity against non-fermenting bacteria of genus *Acinetobacter*, including the most relevant clinical specie *A. baumannii*. To find the reasons for this specific activity, since the cationic antibacterial agents act mainly on bacterial membranes, we have speculated on the structure of the OM of the genus *Acinetobacter* and particularly that of the species *A. baumannii*.

In this regard, it has been reported that *A. baumannii* MDR isolates, unlike *E. coli*, produce hepta-acylated lipid A as their main lipid A species, in contrast to the common hexa-acylated lipid A [62]. In addition, *A. baumannii* lacks the classic long O-polysaccharide antigen but displays an extended oligosaccharide *core* (this different molecule is in fact termed lipo-oligosaccharide (LOS) [63]. Lipids A and LOS *core* parts in *A. baumannii* are phosphorylated to varying extent, generating an overall negative charge [64,65].

When the net negative charge is perfectly stabilized by bivalent cations, the intact OM of *A. baumannii* provides an intrinsic barrier to many potentially toxic compounds, but the unique composition of the OM also makes the cell highly susceptible to cationic materials, which are highly attracted to the strong anionic lipid A of this species. We therefore hypothesized that, G5-PDK possesses the ability to interact stably with the OM of *A. baumannii* and not with that of *E. coli* by virtue of the aforementioned differences.

Furthermore, although experiments to assess the cytotoxic effects of G5-PDK on eukaryotic cells are certainly necessary to support its future pharmaceutical development and are currently underway, due to its structural similarities with the recently reported non-cytotoxic dendrimers [28], we can already hypothesize its non-cytotoxicity on mammalian cells.

## 3. Materials and Methods

### 3.1. Chemicals and Instruments

The uncharged fifth generation polyester-based inner scaffold of G5-PDK (G5-PD-OH) was prepared starting from the AB_2_ monomer, known as *bis*-hydroxymethyl propanoic acid (*bis*-HMPA), performing previously reported procedures [32,33,34,35,36,66,67]. A stylized representation of the structure of G5-PD-OH is available in Section 2. FTIR, NMR data, and elemental analysis results are reported in Section 3.2. All reagents and solvents were purchased from Merck (formerly Sigma-Aldrich, Darmstadt, Germany) and were purified by standard procedures. The *N,N*-*di*-tert- butoxycarbonyl-lysine (Boc_2_-Lys-OH) was purchased by Merck (formerly Sigma-Aldrich, Darmstadt, Germany) and was used as such without further purification. Melting points and boiling points are uncorrected. FTIR spectra were recorded as films or KBr pellets on a Spectrum Two FT-IR Spectrometer (PerkinElmer, Inc., Waltham, MA, USA). ^1^H and ^13^C NMR spectra of all compounds were acquired on a JEOL 400 MHz spectrometer (JEOL USA, Inc., Peabody, MA, USA) at 400 and 100 MHz, respectively. Fully decoupled ^13^C NMR spectra were reported. Chemical shifts were reported in ppm (parts per million) units relative to the internal standard tetramethylsilane (TMS = 0.00 ppm), and the splitting patterns were described as follows: s (singlet), d (doublet), t (triplet), q (quartet), m (multiplet), and br (broad signal). Centrifugations were performed on an ALC 4236-V1D centrifuge at 3400–3500 rpm. Elemental analyses were performed with an EA1110 Elemental Analyzer (Fison Instruments Ltd., Farnborough, Hampshire, England).

Column chromatographies were performed on Merck silica gel (70–230 mesh). Dynamic Light Scattering (DLS) and Z-potential (ζ-p) determinations were performed using a Malvern Nano ZS90 light scattering apparatus (Malvern Instruments Ltd., Worcestershire, UK). A fraction of G5-PDK was also lyophilized using a freeze–dry system (Labconco, Kansas City, MI, USA). A thin layer chromatography (TLC) system employed aluminum-backed silica gel plates (Merck DC-Alufolien Kieselgel 60 F254, Merck, Washington, DC, USA), and detection of spots was made by UV light (254 nm), using a Handheld UV Lamp, LW/SW, 6W, UVGL-58 (Science Company^®^, Lakewood, CO, USA). Organic solutions were dried over anhydrous magnesium sulfate and were evaporated using a rotatory evaporator operating at a reduced pressure of about 10–20 mmHg.

### 3.2. FTIR, NMR Spectral Data, and Elemental Analysis Results of G5-PD-OH

G5-PD-OH. FTIR (KBr, cm^−1^): 3436 (OH), 2936, 1737 (C=OO) [36]. ^1^H NMR (400 MHz, DMSO-*d*_6_), δ (ppm): 1.01, 1.16, 1.18, 1.23, 1.34 (five s signals, 186H, CH_3_ of generations), 1.70 (m, 2H, CH_2_ propanediol), 3.52 (dd, 128H, CH_2_OH), 3.56 (partially overlapped signal, 2H, CH_2_O propanediol), 3.98 (partially overlapped signal, 2H, CH_2_O propanediol), 4.08–4.18 (m, 120H, CH_2_O of four generations), 4.37 (br s, 64H, OH). ^13^C NMR (100 MHz, DMSO-*d*_6_) δ (ppm): 173.94, 171.73 (C=O), 64.27, 63.55 (CH_2_O), 50.13 (quaternary C of fifth generation), 46.12 (other generations detectable quaternary C), 17.05, 16.61 (CH_3_ of generations); found: C, 51.71; H, 7.01. C_313_H_504_O_188_ requires C, 51.67; H, 6.98%.

### 3.3. Synthesis of Lysine-Modified Cationic Dendrimer G5-PDK (128 HCl)

#### 3.3.1. Synthesis of Lysine-Modified Boc-Protected Dendrimer G5-PD-BK

A solution of G5-PD-OH (68.3 mg; 0.0094 mmol) in dry DMF (1.5 mL) was added with Boc_2_-Lys-OH (76.8 equivalent; 250.0 mg; 0.7209 mmol), 4-dimethylaminopyridine (DMAP) (38.4 equivalents; 44.1 mg; 0.3610), and *N*-ethyl-*N*-(3-dimethylamino)propyl carbodiimide hydrochloride (EDC) (76.8 equivalents; 111.9 mg; 0.7209 mmol). The solution was kept under magnetic stirring at r.t. for 24 h then added with 15 mL of ethyl acetate (EtOAc) to produce a suspension which was washed with 10% aq. KHSO_4_ (3 5× 15 mL). The aqueous washings were extracted with EtOAc and the combined organic phases were washed with aq. 15% NaOH followed by water then dried on MgSO_4_ overnight. The removal of the solvent at reduced pressure afforded the Boc-protected lysine-modified dendrimer G5-PD-BK as off white glassy solid (217.5 mg; 0.0077 mmol; 81.8% yield).

FTIR (KBr, cm^−1^): 3380 (NH), 1747 (C=O ester), 1710 (C=O urethane), 1527 (NH). ^1^H NMR (400 MHz, CDCl_3_), δ (ppm): 0.95–1.90 (m, 572H, CH_3_ of dendrimer + CH_2_O propanediol + CH_2_CH_2_CH_2_ of Lys), 1.43 (s, 576H, CH_3_ of Boc), 1.44 (s, 576H, CH_3_ of Boc), 3.10 (m, 128H, CH_2_NH of Lys), 3.56 (partially overlapped signal, 2H, CH_2_O propanediol), 4.25 (m, 314H, CH_2_O of dendrimer and of propanediol + CHNH of Lys), 4.70–5.50 (m, 128H, ^α^NHBoc + ^ε^NHBoc of Lys). ^13^C NMR (CDCl_3_, 100 MHz), δ (ppm): 14.20–17.90 (CH_3_ of G1, G2, G3, G4, G5), 22.57 (CH_2_), 28.36 (CH_3_ of Boc), 28.47 (CH_3_ of Boc), 29.57 (CH_2_), 31.84 (CH_2_), 40.04 (CH_2_NH), 46.42 (quaternary C), 53.37 (CHNH), 65.41–65.60 (CH_2_O of G1, G2, G3, G4, G5), 79.02 (quaternary C of Boc), 79.80 (quaternary C of Boc), 155.63 (C=O urethane), 156.17 (C=O urethane), 172.32 (C=O amino acid + C=O ester of G1, G2, G3, G4, G5), CH_2_ of propanediol not detectable. Found: C, 56.78; H, 8.30; N, 6.00. C_1337_H_2296_N_128_O_508_ requires C, 56.76; H, 8.18; N, 6.34%.

#### 3.3.2. Acidic Deprotection of G5-PD-BK to Obtain G5-PDK * 128 HCl

A solution of G5-PD-BK (208.8 mg; 0.0074 mmol) in methanol (1 mL) was cooled to 0 °C and treated with acetyl chloride (2 equivalents/Boc-groups to be removed; 148.3 mg; 1.8893 mmol; 134.8 µL). The solution was kept at r.t. under magnetic stirring for 24 h, and then it was concentrated at reduced pressure, taken with MeOH, and precipitated into acetone. The dendrimer in the form of hydrochloride was recovered as oil after centrifugation, washed repeatedly with fresh acetone, recovered all times by centrifugation, and finally dried at reduced pressure. G5-PDK * 128 HCl was obtained as white highly hygroscopic solid, which was stored under vacuum over P_2_O_5_. (147.2 mg, 0.0073 mmol, 99% yield).

FTIR (KBr, cm^−1^): 3431 (NH_3_^+^), 1744 (C=O ester), 1635 (NH). ^1^H NMR (400 MHz, DMSO-*d*_6_), δ (ppm): 1.03–1.99 (m, 570H, CH_3_ of dendrimer + CH_2_CH_2_CH_2_ of Lys), 1.70 (m, 2H, CH_2_ propanediol), 2.76 (m, 128H, CH_2_NH_3_^+^ of Lys), 3.56 (partially overlapped signal, 2H, CH_2_O propanediol), 3.99 (m, 64H, CHNH_3_^+^ of Lys), 4.10–4.50 (m, 250H, CH_2_O of propanediol and of dendrimer + CHNH_3_^+^ of Lys), 8.20 (br s,192H, ^α^NH_3_^+^), 8.82 (br s, 192H, ^ε^NH_3_+ of Lys). ^13^C NMR (DMSO-*d*_6_, 100 MHz), δ (ppm): 19.33 (CH_3_), 23.14 (CH_2_), 28.01 (CH_2_), 31.01 (CH_2_), 40.02 (CH_2_NH_3_^+^), 47.70 (quaternary C), 53.55 (CHNH_3_^+^), 67.65–67.82 (CH_2_O and of G1, G2, G3, G4), 170.68–173.33 (C=O of amino acid + ester of G1, G2, G3, G4), CH_2_ of propanediol not detectable. Found: C, 41.78; H, 7.30; N, 9.10; Cl, 22.11 C_697_H_1400_N_128_O_252_Cl_128_ requires C, 41.56; H, 7.00; N, 8.90; Cl, 22.53%.

### 3.4. Determination of Experimental MW of G5-PDK by Volumetric Titration

MW of G5-PDK in the form of hydrochloride was obtained by volumetric titration with HClO_4_ in AcOH [37]. Briefly, an exactly weighted sample of the dendrimer (30.4 mg) was dissolved in AcOH (5 mL), treated with 2–4 mL of a solution of mercury acetate (1.5 g) in AcOH (25 mL), added with a few drops of a solution of quinaldine red (100 mg) in AcOH (25 mL), and titrated with a standardized 0.1612 N solution of HClO_4_ in AcOH, prepared as described in the subsequent Section 3.3.1. The very sharp end point was detected by observing the disappearance of the red color and the appearance of a fine white precipitate. Titrations were performed in triplicate. The experimental MW of G5-PDK was computed according to the following formulae:(1)$$MW=\frac{128*Wt}{V*0.1612}\;$$
where 128 was the number of the NH_2_ groups due to the 64 peripheral lysine residues, *Wt* was the weight of G5-PDK (30.4 mg), *V* was the volume of HClO_4_ in acetic acid which were necessary to reach the titration end point, and 0.1612 was the normality (N) of the HClO_4_ acetic solution. The results were reported as the mean of three determinations ± S.D.

#### Preparation of the Standard Solution of HClO_4_ in AcOH

Acetic anhydride (3 mL) was added to a solution of HClO_4_ 70% (1.4 mL) in AcOH (80 mL), obtaining a colorless solution which was left under stirring at room temperature overnight. The clear yellow solution was made up to 100 mL with AcOH and standardized with potassium acid phthalate. Standardizations were performed in triplicate and results were reported as mean ± S.D. The title of solution was found to be 0.1612 N ± 0.0001.

### 3.5. Potentiometric Titration of G5-PDK

A potentiometric titration was performed at room temperature with a Hanna Microprocessor Bench pH Meter (Hanna Instruments Italia srl, Ronchi di Villafranca Padovana, Padova, Italy), to build up the titration curve of G5-PDK hydrochloride. The dendrimer (30 mg) was dissolved in 30 mL of Milli-Q water (mQ), then was treated with standard 0.1 N NaOH (1.5 mL, pH = 12). The solution was potentiometrically titrated by adding 0.2 mL aliquots of standard 0.1 N HCl up to total 3.0 mL and measuring the corresponding pH values [38]. Titration were performed in triplicate and results were reported as mean ± S.D.

### 3.6. Dynamic Light Scattering (DLS) Analysis

Particle size (in nm), polydispersity index (PDI) and zeta potential (ζ-p) (mV) of G5-PDK were measured at 25 °C, at a scattering angle of 90° in m-Q water by using a Malvern Nano ZS90 light scattering apparatus (Malvern Instruments Ltd., Worcestershire, UK).

Solutions of G5-PDK in m-Q water were diluted to final concentrations to have 250–600 kcps. ζ-p value of G5-PDK was recorded with the same apparatus. The results from these experiments were presented as the mean of three different determinations ± S.D. Concerning the particle size distribution—both the intensity-based results and the number-based results were reported.

### 3.7. Microbiology

#### 3.7.1. Microorganisms

A total of 19 isolates belonging to Gram-positive and Gram-negative species were used in this study. All were clinical strains belonging to a collection obtained from the School of Medicine and Pharmacy of University of Genova, and identified by VITEK^®^ 2 (bioMerieux, Firenze, Italy) or matrix-assisted laser desorption/ionization time-of-flight (MALDI-TOF) mass spectrometric technique (bioMerieux, Firenze, Italy). Of the four tested Gram-positive organisms, two strains belonged to the *Enterococcus* genus, (one *E. faecalis* resistant to vancomycin (VRE) and one *E. faecium* (VRE), 2 strains pertained to the *Staphylococcus* genus, including one methicillin resistant *S. aureus* (MRSA), and one methicillin resistant *S. epidermidis* (MRSE). Regarding the 15 Gram-negative isolates, two strains belonged to *Enterobacteriaceae* family, including two group A carbapenemase-producing isolates, i.e., one *E. coli* and one *K. pneumoniae*. Twelve strains belonged to the non-fermenting *Acinetobacter* genus: six *A. baumannii*, two *A. pittii*, one *A. junii*, one *A. johnsonii*, two *A. ursingii*.

#### 3.7.2. Determination of the MIC

To investigate the antimicrobial activity of G5-PDK on the 19 pathogens, their Minimal Inhibitory Concentrations (MICs) were determined by following the microdilution procedures detailed by the European Committee on Antimicrobial Susceptibility Testing EUCAST [68].

Briefly, overnight cultures of bacteria were diluted to yield a standardized inoculum of 1.5 × 10^8^ CFU/mL. Aliquots of each suspension were added to 96-well microplates containing the same volumes of serial 2-fold dilutions (ranging from 1 to 512 μg/mL) of G5-PDK to yield a final concentration of about 5 × 10^5^ cells/mL. The plates were then incubated at 37 °C. After 24 h of incubation at 37 °C, the lowest concentration of G5-PDK that prevented a visible growth was recorded as the MIC. All MICs were obtained in triplicate, the degree of concordance in all the experiments was 3/3, and the S.D. was zero.

#### 3.7.3. Killing Curves

Killing curve assays for G5-PDK were performed on representative isolates of *A. baumannii*, *A. pittii*, and *A. ursingii* as previously reported [69]. Experiments were performed over 24 h at G5-PDK concentrations of four times the MIC for all strains.

A mid logarithmic phase culture was diluted in Mueller–Hinton (MH) broth (Merck, Darmstadt, Germany) (10 mL) containing 4 × MIC of the selected compound to give a final inoculum of 1.0 × 10^5^ CFU/mL. The same inoculum was added to MB broth and run in parallel, as a growth control, tubes were incubated at 37 °C with constant shaking for 24 h. Samples of 0.20 mL from each tube were removed at 0, 1, 2, 4, 6, 8, and 24 h, diluted appropriately with a 0.9% sodium chloride solution to avoid carryover of G5-PDK being tested, plated onto MH plates, and incubated for 24 h at 37 °C. Growth controls were run in parallel. The percentage of surviving bacterial cells was determined for each sampling time by comparing colony counts with those of standard dilutions of the growth control. Results have been expressed as log10 of viable cell numbers (CFU/mL) of surviving bacterial cells over a 24 h period. Bactericidal effect was defined as a 3 log10 decrease of CFU/mL (99.9% killing) of the initial inoculum. All time-kill curve experiments were performed in triplicate.

## 4. Conclusions

In this study, a new self-biodegradable fifth generation dendrimer, characterized by an uncharged hydrophobic inner matrix and a lysine-modified highly cationic surface, has been synthetized and characterized. Its structure was validated with FTIR, NMR experiments and elemental analyses, while its MW was estimated by ^1^H NMR experiments, volumetric titrations and was confirmed by elemental analyses. Results from DLS analysis established that G5-PDK has a particle size and a surface charge suitable for effective biomedical applications. Interestingly, microbiologic results recognized that G5-PDK possesses promising antibacterial in vitro activity against MDR bacteria of genus *Acinetobacter* and particularly against *A. baumannii*. In addition, G5-PDK, dissimilar from other cationic molecules present in nature, such as AMPs, has shown to possess a unique cationic amphiphilic structure, characterized by very low MIC values and powerful bactericidal behavior.

The main merit of this study consists of having achieved very fascinating advances in the fight against this almost intractable bacterial species, in a scenario of scarce data regarding the activity of newly designed cationic dendrimers, against *A. baumannii*. More importantly, unlike most studies, only clinical isolates were considered in this work.

## Data Availability

All data concerning this study are contained in the present manuscript or in previous articles (references have been provided).

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
