# Peer review of "Bactericidal Activity of a Self-Biodegradable Lysine-Containing Dendrimer against Clinical Isolates of Acinetobacter Genus"

_ijms, 2021, doi:10.3390/ijms22147274_

Round 1
Reviewer 1 Report
The manuscript entitled: „Bactericidal Activity of a Self-biodegradable Lysine-containing Dendrimer Against Clinical Isolates of Acinetobacter Genus Silvana Alfei“ presents the synthesis of a new dendrimer named G5-PDK and its activity against clinical isolates of A. baumannii, and on other isolates of the genus Acinetobacter. Time-kill studies were also performed on representative MDR strains of A. baumannii and on isolates of A. pittii and A. ursingii.
Firstly, the authors reported on the pathways used to obtain and characterize (NMR, elemental analyses, FTIR, volumetric titration, DLS) the final cationic dendrimer G5-PDK. Then, there are presented the results obtained for the screening of the antibacterial activity against both Gram-positive and Gram-negative species, including MDR isolates. Also, the time-kill experiments were performed with G5-PDK at concentrations equal to 4 x MIC on selected strains of A. baumanni, A. pittii, and A. ursingii.
In general, the data are strong and convincingly show the necessity to develop new strategies to obtain antibacterial compounds. This manuscript is well written and concise, and the data obtained to sustain the conclusions.
Please add in the Abstract more information about the compound obtained and tested besides its abbreviation (G5-PDK: new fifth-generation lysine-modified dendrimer) and the info about the compound named G5K. Also included in the Abstract experimental values obtained in this study (e.g. antibacterial activity, values of MIC).
Please remove the reference (line 748) from the Conclusions and move this observation to the Discussion part. It is preferred that this part should contain your conclusions.
Author Response
The manuscript entitled: “Bactericidal Activity of a Self-biodegradable Lysine-containing Dendrimer Against Clinical Isolates of Acinetobacter Genus Silvana Alfei” presents the synthesis of a new dendrimer named G5-PDK and its activity against clinical isolates of A. baumannii, and on other isolates of the genus Acinetobacter. Time-kill studies were also performed on representative MDR strains of A. baumannii and on isolates of A. pittii and A. ursingii.
Firstly, the authors reported on the pathways used to obtain and characterize (NMR, elemental analyses, FTIR, volumetric titration, DLS) the final cationic dendrimer G5-PDK. Then, there are presented the results obtained for the screening of the antibacterial activity against both Gram-positive and Gram-negative species, including MDR isolates. Also, the time-kill experiments were performed with G5-PDK at concentrations equal to 4 x MIC on selected strains of A. baumanni, A. pittii, and A. ursingii.
In general, the data are strong and convincingly show the necessity to develop new strategies to obtain antibacterial compounds. This manuscript is well written and concise, and the data obtained to sustain the conclusions.
We thank the Reviewer for his positive general comments concerning our study.
Please add in the Abstract more information about the compound obtained and tested besides its abbreviation (G5-PDK: new fifth-generation lysine-modified dendrimer) and the info about the compound named G5K. Also included in the Abstract experimental values obtained in this study (e.g. antibacterial activity, values of MIC).
The information requested have been added in the abstract. Please, see lines 18, 20-21 and 22-25.
Please remove the reference (line 748) from the Conclusions and move this observation to the Discussion part. It is preferred that this part should contain your conclusions.
As requested, the reference and the related observation have been removed from the Conclusions section. The observation has been inserted in the Discussion part. Particularly, it has been moved in Section 2.6.4., lines 569-573.
Reviewer 2 Report
The article is interesingly presented.
Remarks to Table 3 - MIC values for Ciprofloxacin as the reference drug should be added for all strains. Table 4 - MIC values for Ciprofloxacin should be added separately for each strain.
Author Response
The article is interestingly presented.
Remarks to Table 3 - MIC values for Ciprofloxacin as the reference drug should be added for all strains. Table 4 - MIC values for Ciprofloxacin should be added separately for each strain.
As required, separated MIC values for each strain were entered for ciprofloxacin in both Table 3 and Table 4. As a result, the main text has been slightly modified (lines 398-400).